# Four Novel Mycoviruses from the Hypovirulent *Botrytis cinerea* SZ-2-3y Isolate from *Paris polyphylla*: Molecular Characterisation and Mitoviral Sequence Transboundary Entry into Plants

**DOI:** 10.3390/v14010151

**Published:** 2022-01-14

**Authors:** Qiong Wang, Qi Zou, Zhaoji Dai, Ni Hong, Guoping Wang, Liping Wang

**Affiliations:** 1College of Plant Science and Technology, Huazhong Agricultural University, Wuhan 430070, China; wq17737199887@163.com (Q.W.); 13613570619@163.com (Q.Z.); whni@mail.hzau.edu.cn (N.H.); gpwang@mail.hzau.edu.cn (G.W.); 2Key Laboratory of Plant Pathology of Hubei Province, Huazhong Agricultural University, Wuhan 430070, China; 3Key Laboratory of Green Prevention and Control of Tropical Plant Diseases and Pests, College of Plant Protection, Hainan University, Ministry of Education, Haikou 570100, China; daizhaoji1020@126.com

**Keywords:** *Botrytis cinerea*, mycovirus, *Botourmiaviridae*, *Mitovirus*, viral genome, coinfection, hypovirulence, *Paris polyphylla*

## Abstract

A hypovirulent SZ-2-3y strain isolated from diseased *Paris polyphylla* was identified as *Botrytis cinerea*. Interestingly, SZ-2-3y was coinfected with a mitovirus, two botouliviruses, and a 3074 nt fusarivirus, designated Botrytis cinerea fusarivirus 8 (BcFV8); it shares an 87.2% sequence identity with the previously identified Botrytis cinerea fusarivirus 6 (BcFV6). The full-length 2945 nt genome sequence of the mitovirus, termed Botrytis cinerea mitovirus 10 (BcMV10), shares a 54% sequence identity with Fusarium boothii mitovirus 1 (FbMV1), and clusters with fungus mitoviruses, plant mitoviruses and plant mitochondria; hence BcMV10 is a new *Mitoviridae* member. The full-length 2759 nt and 2812 nt genome sequences of the other two botouliviruses, named Botrytis cinerea botoulivirus 18 and 19 (BcBoV18 and 19), share a 40% amino acid sequence identity with RNA-dependent RNA polymerase protein (RdRp), and these are new members of the *Botoulivirus* genus of *Botourmiaviridae*. Horizontal transmission analysis showed that BcBoV18, BcBoV19 and BcFV8 are not related to hypovirulence, suggesting that BcMV10 may induce hypovirulence. Intriguingly, a partial BcMV10 sequence was detected in cucumber plants inoculated with SZ-2-3y mycelium or pXT1/BcMV10 agrobacterium. In conclusion, we identified a hypovirulent SZ-2-3y fungal strain from *P. polyphylla*, coinfected with four novel mycoviruses that could serve as potential biocontrol agents. Our findings provide evidence of cross-kingdom mycoviral sequence transmission.

## 1. Introduction

High-throughput sequencing and bioinformatics have revealed the presence of numerous mycoviruses in filamentous fungi, especially mitovirus and ourmia-like viral sequences [1,2,3], and the observed genetic variability has implications for understanding the evolution of viruses in fungi [4,5,6,7]. Mitoviruses and ourmia-like mycoviruses belong to *Mitoviridae* and *Botourmiaviridae*, respectively. Members of these two families are positive-sense single-stranded RNA (+ssRNA) viruses, with genome sequences ranging from 2.2 to 3.6 kb that only encode a single and essential RNA-dependent RNA polymerase protein (RdRp) [8,9,10]. *Botourmiaviridae* includes six genera, in which *Ourmiavirus* infects plants, *Scleroulivirus* infects both fungi and plants, and the other four genera (*Botoulivirus, Magoulivirus*, *Penoulivirus* and *Rhizoulivirus*) mainly infect fungal hosts such as *Botrytis cinerea*, *Sclerotinia sclerotiorum*, *Rhizocotonia solani* and *Magnaporthe*
*o**ryzae*, according to the International Committee on Taxonomy of Viruses (ICTV) released in 2020 (https://talk.ictvonline.org/taxonomy/, accessed on 28 July 2021). Mycoviruses in *Botourmiaviridae* have a wide range of hosts, revealing viral diversity and evolution. Ourmia-like mycoviral sequences closely related to plant ourmiaviruses are widely distributed in eukaryotes, according to database searches [10,11]. Most ourmia-like mycoviruses do not appear to affect fungus hosts such as *Magnaporthe*
*oryzae* or *Pyricularia oryzae* [12,13], with the exception of Fusarium oxysporum ourmia-like virus 1 (FoOuLV-1), which belongs to the genus *Magoulivirus* and exhibits hypovirulence [14].

Members of the *Mitoviridae* family include the *Mitovirus* genera, which replicate mainly in mitochondria [9]. Some mitoviruses, including Botrytis cinerea mitovirus 1 (BcMV1), Sclerotinia sclerotiorum mitovirus 1/HC025 (SsMV1/HC025), SsMV1, and SsMV2/KL-1, have obvious effects on fungi; these include mitochondrial malformation, and loss of function, which result in hypovirulence to hosts [15,16,17]. By contrast, some mitoviruses lead to latent infection in hosts [18]. Mitovirus genomic sequences are predicated to transfer between fungi and plants, most likely due to fungus-mediated horizontal gene transfer (HGT) events between fungi and plants, and among viruses during long-term co-evolution [19,20]. Bioinformatic analysis revealed that mycoviral sequences are present in various organisms, especially plants, and that mitoviruses are widely distributed across kingdoms [17,21]. Recently, Chenopodium quinoa mitovirus 1 was found to infect plant hosts and cause severe symptoms without integration into the plant genome [22,23]. It is necessary to further explore the triadic relationship of plants, fungi and mitoviruses.

Mycoviruses may induce hypovirulence in fungi infecting horticulture and oil crops; some have been applied to disease control, especially Cryphonectria hypovirus 1 (CHV1), Rosellinia necatrix megabirnavirus 1 (RmBV1) and Sclerotinia sclerotiorum hypovirulence-associated DNA virus 1 (SsHADV-1). These mycoviruses effectively control blight disease in chestnut, rosellinia trunk rot disease in apple, and sclerotia disease in oilseed rape, respectively [24,25,26]. Hence, researchers have paid much attention to identifying and characterising different types of mycoviruses as potential and effective biological agents for environmentally friendly control of plant diseases in the field. *Botrytis* species in the family *Sclerotiniaceae* are the main causal agents of grey mold disease affecting agriculturally important crops and various horticultural plants, including strawberry, peach, pear, grape, lettuce, cucumber and tomato, resulting in serious economic losses [27,28,29]. Application of Fungicides is the main method to control grey mold diseases in the short term. However, long-term overuse of fungicides results in environmental pollution, the development of drug resistance, and ineffective control [27,29,30].

By contrast, the biological control of grey mold disease, exemplified by the application of CHV1 and RmBV1 in horticultural crops, appears to be safe and efficient [24,31]. Numerous mycoviruses have been reported to infect *Botrytis* species on horticultural plants. These viruses mainly consist of +ssRNA, including *Hypoviridae* (Botrytis cinerea hypovirus 1) [32], *Mitoviridae* (Botrytis cinerea mitovirus 1, BcMV1) [15], *Endoviridae* (*B. cinerea* endornavirus 1, BcEV1) [33], *Botourmiaviridae* (Botrytis ourmia-like virus) [34], *Fusariviridae* (Botyris cinerea fusarivirus 1) [32], *Gammaflexiviridae* (Botrytis virus F, BVF) [35] and *Alphaflexiviridae* (Botrytis virus X, BVX) [36]. The double-stranded RNA (dsRNA) mycoviruses include *Mymonaviridae* (Botrytis cinerea mymonavirus 1, BcMyV1) [32], *Partitiviridae* (Botrytis cinerea partitivirus 1, BcPV1) [37], Botrytis cinerea partitivirus 2 (BcPV2) [38], Botryotinia fuckeliana partitivirus 1 (BfPV1) [32], *Totiviridae* (Botryotinia fuckeliana totivirus 1, BfTV1) [39,40] and *Botybimaviridae* (Botrytis porri RNA virus 1, BpRV1) [41]. In addition, *Bunyaviridae* with an ssRNA genome (Botrytis cinerea negative-stranded RNA virus 1, BcNSRV-1) [42] and two circular ssDNA (Botrytis gemydayirivirus, BGDaV1 and BGDaV2) [43] have also been reported. Interestingly, among these viruses, some have been identified as being able to weaken the pathogenicity of hosts, including BcMV1, BcHV1, BcPV1, BcPV2, BpRV1, Botrytis cinerea RNA virus 1 (BcRV1), BcMyV1 and BGDaV1. These are potential biocontrol agents for fungal diseases in the field [15,32,37,38,41,43,44].

*Paris polyphylla* is a horticultural crop with high value in traditional Chinese medicines. In June 2018, various *P. polyphylla* plants were killed by grey mold disease in Sui County, Hubei Province, China. The SZ-2-3y strain was isolated from the diseased leaves of *P. polyphylla* and further identified as a *B. cinerea* strain. The SZ-2-3y isolate showed relatively normal morphology, but exhibited hypovirulent characteristics in in vitro pathogenicity tests, including the leaves of *P. polyphylla* and tobacco (*Nicotiana benthamiana*), and the fruits of tomato and apple. By screening viral contigs and dsRNA patterns, and performing RT-PCR, two ourmia-like viruses designated BcBoV18 and BcBoV19, one mitovirus designated BcMV10, and a fusarivirus designated BcFV8, were identified in SZ-2-3y.

To further explore the potential of these mycoviruses for biological control, the molecular characterisation of mycoviruses and their biological functions in the *B. cinerea* strain SZ-2-3y were conducted in the present work. Interestingly, BcMV10 sequences were also detected in the new leaves of cucumber plants inoculated with SZ-2-3y mycelium, as well as agrobacterium-mediated inoculation of pXT1/BcMV10. The findings provide new insight into the introduction of mitovirus sequences from fungi into plants, mediated by HGT.

## 2. Materials and Methods

### 2.1. Isolation of the Pathogenic SZ-2-3y Strain and Detection in P. polyphylla

From April to June 2018, various severely diseased *P. polyphylla* plants with a grey mold layer on their flowers and leaves were identified in Sui County, Hubei, China. The pathogen was isolated from the diseased leaves and flowers. Genomic DNA was extracted from the above isolated mycelium and designated the SZ-2-3y strain, based on the Cetylramethylammonium bromide (CTAB) method, with slight modifications. Internal transcribed space (ITS)-rDNA region sequence analysis, in combination with PCR detection using *B. cinerea*-specific PCR primers, was used to identify *Botrytis* species [45]. Meanwhile, the *B. cinerea* B05.10 isolate was provided by Prof. Li [15].

### 2.2. Characterisation of SZ-2-3y Strain Growth Morphology and Pathogenicity

The isolated SZ-2-3y strain mycelial plugs, collected from actively growing colonies on 2-day-old agar plates, and the B05.10 reference isolate were incubated on potato dextrose agar (PDA) at 20 °C and 25 °C in darkness, to determine radial growth rates, sclerotia production, and quantities over 12 days. The colony diameter was measured by the cross method to calculate the growth rate. For determination of SZ-2-3y pathogenicity, fresh mycelial agar plugs with 5 mm diameter colonies were placed on fully expanded leaves detached from potted *P. polyphylla* and *N. benthamiana*, on tomato fruits from a supermarket by intact inoculation, and on apple fruits (*Malus mali*) from a supermarket by wound inoculation. Meanwhile, the B05.10 reference strain and blank PDA agar inoculation served as controls. The samples were placed in a plastic tray with moist white cotton gauze and a transparent plastic covering to retain humidity, and the diseased lesion diameter was measured at 24 h intervals. The results were analysed by one-way analysis of variance (ANOVA) at a significance level of 0.05 using SPSS software (version 22.0) to determine differences between treatments, and the number of sclerotia and lesion diameters were analysed by Student’s *t* test at *p* < 0.05 and *p* < 0.01, respectively.

### 2.3. DsRNA Extraction and Detection

Strain SZ-2-3y hyphae were cultured on PDA covered with cellophane for 5 to 7 days. The mycelium mass was harvested and quickly ground to a fine powder in liquid nitrogen. Two methods were employed to extract dsRNA: a cross-column method developed in our lab, and a cellulose (CF-11) chromatography method reported previously [46,47]. The obtained dsRNAs were separated by 1.2% agarose gel electrophoresis to visualise patterns, and stored for subsequent experiments.

### 2.4. High-Throughput Sequencing and Analysis

High-throughput sequencing was used to identify novel mycoviruses in fungi. Four mycelium samples from three *Alternaria* spp. strains and SZ-2-3y from the present study were used for Illumina sequencing to screen for mycoviruses. Total RNAs from mixtures of cultured mycelial mass were extracted and further assessed by Nanodrop and Agilent 2100 instruments, and by gel electrophoresis. The rRNA-depleted RNA library was constructed using the Epicentre Ribo-Zero rRNA removal kit (Epicentre, Madison, WI, USA) and a TruSeq RNA Sample Prep Kit v2 (Illumina, San Diego, CA, USA). Sequencing was performed by the Illumina HiSeq X-ten sequencing platform with PE150 bp (Biomarker Technologies, Beijing, China). Clean reads of a high quality were obtained by filtering low-quality raw data, including adapter-contaminated reads, reads with unknown bases (N), and short reads. After quality trimming, de novo assembly was performed with IDBA_ud software. Subsequently, assembled contigs were further analysed to search for homology with mycoviral sequences using BLASTn and BLASTx in the NCBI database. The obtained mycoviral sequences from the library presence were detected and confirmed by RT-PCR for SZ-2-3y in this study (Appendix A).

### 2.5. Full-Length Cloning of Mycovirus cDNA

Intermediate fragments of mycoviruses were obtained by reverse transcription PCR (RT-PCR) using virus-specific primers, designed based on the assembled mycovirus contig sequences. The 5′-terminal and 3′-terminal sequences of dsRNA were determined as previously described [48]. The adaptor primer RACE-OLIGO was ligated to the 3′-terminus of each dsRNA strand using T4 RNA ligase (New England BioLabs, Beijing, China) at 25 °C, for 2 h. The adaptor-ligated dsRNAs were purified, reverse transcribed using O5RACE-1 primer (the RT primer complementary to the adaptor primer RACE-OLIGO sequence). Synthetic cDNAs were amplified using primers O5RACE-2 or O5RACE-3, complementary to the RNA ligation oligonucleotide and mycovirus sequence-specific primers corresponding to the 5′- and 3′-terminal sequences of dsRNAs, to perform amplification of viral cDNA ends. The experiment was repeated at least three times independently. Primers used in this study are listed in Appendix A. All PCR products were purified and ligated into the pMD18-T vector (Takara, Dalian, China), then transformed into *Escherichia coli* Top 10 competent cells for DNA sequencing.

### 2.6. Analysis of Mycoviral Genome Sequences

The sequences obtained from clones were assembled using DNAMAN. The viral open reading frames (ORFs) were predicted using ORF finder (https://www.ncbi.nlm.nih.gov/orffinder, accessed on 28 July 2021). The mycoviral sequences were searched for using the NCBI database. Multiple mycoviral sequences were aligned using MAFFT (https://mafft.cbrc.jp/alignment/server/index.html, accessed on 28 July 2021). The phylogenetic analysis was conducted via neighbour-joining (NJ) and maximum likelihood (ML) methods with 1000 replicates using MEGA 5.2, and the RdRp motif sequences were analysed using GeneDoc software [49,50]. The bootstrap values (>50%) are labelled on branches. The information for the selected mycoviral isolates used in the phylogenetic tree and multiple alignment comparisons of RdRp amino acid (aa) sequences is listed in Appendix A. Potential secondary structures of terminus sequences from the three mycovirus genomes were predicated by RNAfold (http://mfold.rna.albany.edu/?q=DINAMelt/Quickfold, accessed on 28 July 2021) [51]. The resulting RNA reads were mapped to mycoviral genomes using HISAT version 2.22 [52]. The number of mapped reads for each mycovirus was calculated by Samtools [53], and duplicates were identified using Picard version 2.2.4 (https://broadinstitute.github.io/picard/, accessed on 28 July 2021). The sequencing depth of viral sites was calculated and visualised using Circos software.

### 2.7. Horizontal Transmission of Mycoviruses in SZ-2-3y

The dsRNA-containing hypovirulent strain SZ-2-3y, as the donor, was horizontally transmitted to the virulent isolate B05.10, as the recipient, on PDA using the pair-culturing technique. From colonies of each recipient in 7-day-old cultures, six derivatives (Yb-1, Yb-2, Yb-3, Yb-7, Yb-8 and Yb-11) were tested for the presence of the four mycoviruses by RT-PCR with mycovirus-specific primer pairs (Appendix A). Additionally, these derivatives were tested for biological characteristics, namely the growth rate and phenotype on PDA, and for pathogenicity by intact inoculation on tobacco leaves using the above-described methods. Parental SZ-2-3y and B05.10 strains were included as controls.

### 2.8. SZ-2-3y Mycelium Inoculation on Plants

For inoculation of the SZ-2-3y strain to cucumber plants, mycelia-containing PDA plugs, picked up from the edge of 2- or 3-day-old culture colonies with active infection ability, were placed on the cotyledons of cucumber plants. The inoculated parts of the leaves were wrapped with plastic wrap for 48 h. The healthy cucumber plants and plants inoculated with blank PDA or B05.10 served as controls. The inoculated plants were grown in pots under a 16 h light/8 h dark cycle at 24 ± 2 °C, in a growth chamber. Two or three independent experiments were conducted. Cucumber seeds from the ‘Suyo’ cultivar (cv) were provided by Prof. Shifang Li from the Chinese Academy of Agricultural Science (CAAS), and seeds of the ‘Cuiyu’ cultivar were bought from online supermarkets.

### 2.9. Construction of the pXT1/BcMV10 Vector

The modified binary plant expression vector pXT1 (designated as pCB301-2 × 35S-MCS-HDVRZ-NOS), derived from the original vector pCB301 and used as the backbone for the construction of infectious clones, was kindly supplied by Prof. Xiaorong Tao (Nanjing Agricultural University) [54]. The homologous recombination method was employed for construction [55,56]. The full-length BcMV10 cDNA was amplified, including the 30 nt vector sequence; the 5′-end of the first viral genome fragment was joined through a 15 nt overlap to the end of the 35 S promoter of pXT1, and the 3′-end viral fragment was joined through a 15 nt overlap to the 5′ end of the HDV ribozyme of pXT1 (Appendix A). The pXT1 plasmid was linearised with restriction enzymes *Stu*I and *Sma*I (Takara, Dalian, China). The resulting PCR fragments and linear pXT1 were purified to generate the full-length BcMV10 clone using a ClonExpress II One Step Cloning Kit (Vazyme Biotech Co. Ltd., Nanjing, China) according to the manufacturer’s procedure, with slight modifications. The obtained assembled DNA products were transformed into *E. coli* strain Top10, and positive colonies were confirmed by DNA sequencing. The resulting recombinant plasmid (named pXT1/BcMV10) was used to transform *Agrobacterium tumefaciens* GV3101 by heat-shock transformation for infiltration experiments [57].

### 2.10. Ago-Infiltration of Cucumber Plants with pXT1/BcMV10

The obtained positive plasmids designated pXT1/BcMV10, and the pXT1 control, were separately transformed into GV3101, and the *Agrobacteria* cells were grown at 28 °C for 2 days on LB solid medium with 50 μg mL^−1^ kanamycin and 25 μg mL^−1^ rifampicin antibiotics. The positive monoclonal culture was shaken at 28 °C overnight, and cells were washed with sterile water three times after centrifugation. The culture was suspended in inoculation buffer including 0.1 M MgCl_2_, 0.1 M 2-Morpholineethanesulfonic acid (MES), 0.015 M Acetosyringone (AS) and the cell density was adjusted to an OD600 value of 1.0. The culture was incubated for 4 h at room temperature then used for inoculation. The cultures (300 μL) were agroinfiltrated into the abaxial surface of two cotyledons from ‘Cuiyu’ and ‘Suyo’ seedlings, using 1 mL syringes without a needle. The inoculated plants were grown under 16 h/8 h light/dark conditions at 24 °C. Two or three independent experiments were conducted.

### 2.11. Total RNA Extraction and RT-PCR Analysis of Inoculated Plants

At about 2 weeks post-inoculation (dpi), the leaves and roots of cucumber shoots were evaluated to detect mycoviral sequences by RT-PCR using viral-specific primer pairs (Appendix A). Total RNAs were extracted using TRIzol (Aidlab Biotechnologies Co., Ltd., Beijing, China) from: the local and new leaves or roots of cucumber plants, inoculated with pXT1/BcMV10 or SZ-2-3y mycelium; new leaves of control plants, inoculated with PDA or pXT1; and healthy cucumber plants as negative controls. cDNAs were synthesised by M-MLV reverse transcriptase (Promega, Madison, WI, USA). BcMV10 sequence-specific primers and 2× DNA polymerase mixture (ComWin Biotech Co., Ltd., Beijing, China) were used for RT-PCR, and the resulting PCR products were verified by gel electrophoresis analysis and DNA sequencing.

## 3. Results

### 3.1. Identification and Biological Characteristics of the Hypovirulent SZ-2-3y Strain

In June 2018, some severely diseased *P. polyphylla* plants were identified in Sui County, Hubei, China, with obvious grey mold symptoms on their leaves and flowers (Appendix A). The SZ-2-3y strain was isolated from *P. polyphylla* leaves with typical grey mold disease, and identified based on cultural morphology and PCR detection. The rDNA-ITS nucleotide sequence from the mycelium of the SZ-2-3y strain was 99% identical to that of the *B. cinerea* strains. Meanwhile, a 327 bp specific DNA fragment was amplified using reported *B. cinerea*-specific primers Bc-f/Bc-r and sequenced, revealing that the SZ-2-3y isolate was indeed *B. cinerea* (Appendix A). Compared with B05.10 controls—with 1.9 cm/day after incubation at 20 °C, for 5 days, in darkness—the colony morphology of the SZ-2-3y strain was normal, with a slightly slower growth rate of 1.6 cm/day (Figure 1A,B); furthermore, more sclerotia were produced (143/plate compared with 50/plate for B05.10) after 12 days (Figure 1C). Upon incubation at 25 °C for 4 days, in darkness, the colony morphology of the SZ-2-3y strain was abnormal, mycelia were sparse, and the diameter of colonies was smaller, with an average growth rate of 0.8 cm/day compared with 1.9 cm/day for B05.10. Thus, temperature had a significant effect on SZ-2-3y morphology.

Pathogenicity tests were also conducted to investigate the virulence of SZ-2-3y. No lesions were observed on plants that were inoculated with the SZ-2-3y strain and grown at room temperature after 4 dpi, regardless of the inoculation method (intact inoculation for *P. polyphylla*, tobacco leaves and tomato fruits, or wound inoculation for apple fruits). By contrast, B05.10 caused large necrotic lesions on the leaves of *P. polyphylla* and tobacco, and on the fruits of tomato and apple, with average lesion diameters of 0.86, 2.56, 2.54 and 3.28 cm, respectively (Figure 1D,E). These results reveal that the SZ-2-3y strain was hypovirulent.

### 3.2. Mixed Mycovirus Infection with the SZ-2-3y Strain

#### 3.2.1. Identification of Four Mycoviruses in the SZ-2-3y Strain

Since the SZ-2-3y strain exhibited hypovirulence and apparent phenotypic changes under the influence of temperature, we questioned whether it could infect mycoviruses, and if mycoviruses were responsible for the hypovirulence of the SZ-2-3y strain. We first extracted dsRNAs from SZ-2-3y mycelia, and obtained bands corresponding to fragments ~8 kb and ~3 kb in length following SI nuclease (Thermo Fisher Scientific, Waltham, MA, USA) digestion or long-time storage, to trigger rRNA degradation (Appendix A). To identify the sequences amplified from the SZ-2-3y strain, high-throughput sequencing was carried out. The obtained high-quality reads were further assembled, and obtained contigs were aligned using BLASTx in NCBI. To clarify whether SZ-2-3y carries mycoviruses, RT-PCR was performed using total RNAs or dsRNA extracted from the SZ-2-3y strain as template, in combination with specific primers designed from the obtained contigs. RT-PCR results and sequencing analysis resulted in 587 nt, 578 nt, 523 nt and 260 nt fragments, in accordance with the four mycovirus contigs obtained by high-throughput sequencing, which were confirmed to be derived from the SZ-2-3y strain (Appendix A). We obtained six mycoviral contigs (Contig111, Contig334, Contig420, Contig25026, Contig12923 and Contig46148; Appendix A). A homology search using BLASTx showed that RdRps of the four novel mycoviruses were most closely related to the homolog in Fusarium boothii mitovirus 1 (FbMV1; 54%), Sclerotinia sclerotiorum ourmia-like virus 13 (79%), Botrytis cinerea ourmia-like virus 12 (79%) and Botrytis cinerea fusarivirus 5 (85%), as shown in Table 1. Since nine mitoviruses in BcMV1-9 have been identified in *B. cinerea* and grape, the mitovirus found in the SZ-2-3y strain was designated as Botrytis cinerea mitovirus 10 (BcMV10). Meanwhile, 17 Botrytis cinerea ourmia-like viruses have been identified in *B. cinerea* and reported in the NCBI database and the literature. The botouliviruses identified in SZ-2-3y were designated Botrytis cinerea botoulivirus 18 (BcBoV18) and Botrytis cinerea botoulivirus 19 (BcBoV19), based on ICTV classification (https://talk.ictvonline.org/taxonomy/, accessed on 28 July 2021).

#### 3.2.2. Analysis of the Fusarivirus Genomic Sequence

Seven fusariviruses have been found in *B. cinerea* strains, and the fusarivirus identified in this study was designated Botrytis cinerea fusarivirus 8 (BcFV8). The obtained 3074 nt sequence of BcFV8 has been submitted to NCBI under GenBank accession number OL321741 (Appendix A). BcFV8 encodes a partial RdRp sequence that shares the highest identity (87.2%) with its homolog in BcFV6, and it clusters in the same group as other members in the *Fusarivirus* genus (Appendix A, Figure 2).

### 3.3. Analysis of Full-Length cDNA Genome Sequences: A Novel Mitovirus and Two New Botouliviruses

The sequences of full-length cDNAs were determined by assembling partial cDNAs amplified from the purified dsRNAs or tRNAs as templates, using RT-PCR with random primers and cDNA end amplification with adaptor primers. RT-PCR, combined with rapid amplification of cDNA ends (RACE) cloning and sequencing, yielded intermediate sequences of three viruses (2452 nt for BcMV10, 1899 and 935 nt for BcBoV18, and 1731 and 1342 nt for BcBoV19) from specific designed primers. The obtained 5′- and 3′- RACE sequences were 452 and 301 nt for BcMV10, 202 and 287 nt for BcBoV18, and 387 and 192 nt for BcBoV19 (Appendix A). The obtained full-length genome sequences have been deposited in GenBank under accession number OK634394 for BcMV10, OK634395 for BcBoV18 and OK634396 for BcBoV19 (Appendix A).

#### 3.3.1. A Novel Mitovirus

The genomic structure of the mitovirus is shown in Figure 3A. The full-length genome sequence of BcMV10 is 2945 bp with an AU-rich element, and it contains a single ORF, spanning positions 361 to 2817 nt, which is predicted to encode an RdRp protein of 818 aa. The 5′- untranslated region (UTR) and 3′- UTR are 360 nt and 128 nt, respectively. The highest RdRp aa sequence identity of BcMV10 is 54%, shared with FbMV1; this is followed by other mitoviruses in fungi hosts, including Nigrosp oraoryzae mitovirus 1 (NoMV1; 50%); other plant mitoviruses, including Solanum chacoense mitovirus 1 (19%, 115/591 RdRp aa); and plant mitochondria, including *Solanum tuberosum* (24%, 74/310 RdRp aa) (Appendix A). Both the 5′- and 3′- UTRs of the BcMV10 genome are clearly folded into two potential stem-loop structures. In addition, the long panhandle structure is attached to the 3′- and 5′-termini sequences through base pairing (Appendix A).

#### 3.3.2. Two Novel Botouliviruses

The full-length genome sequences of BcBoV18 and BcBoV19 are 2759 nt and 2812, respectively, and they possess a single ORF encoding RdRp of 1986 nt and 2085 nt, respectively (Figure 3B,C). The genomes of BcBoV18 and BcBoV19 contain AUs with 51.3% and 56% identity, respectively. The RdRp encoded by the ORF of BcBoV18 is most closely related to that of BcOlV17, while that of BcBoV19 is most closely related to that of Botrytis cinerea ourmia-like virus 12 (BcOlV12), with RdRp aa identities of 78%, followed by members of the *Botoulivirus* genus (Appendix A). The 5′-UTR and 3′-UTR are 43 nt and 730 nt for BcBoV18, and 54 nt and 673 nt for BcBoV19. The 5′- and 3′-UTRs of BcBoV18 and BcBoV19 genomes could form potential terminal stable stem-loop structures (Appendix A).

### 3.4. Molecular Phylogenetic Analysis of Viral RdRps from BcMV10, BcBoV18, BcBoV19 and Their Relatives

A phylogenetic tree and multiple alignment analyses of the RdRp sequences were performed on the three newly discovered mycoviruses, 25 representative members belonging to *Mitoviridae*, and 24 representative members in *Botourmiaviridae* from the NCBI database.

#### 3.4.1. A Novel Mitovirus

We searched for the presence of BcMV10-like sequences in the NCBI database using the putative BcMV10-encoded protein as the query, revealing several significant matches, mainly in fungi associated with plants (Appendix A). The phylogenetic tree indicated two subgroups supported by high bootstrap values. One subgroup includes fungi mitoviruses consisting of three clades (Clades I, II and III); another subgroup contains plant mitoviruses such as Solanum chacoense mitovirus 1, Chenopodium quinoa mitovirus 1 and Cannabis sativa mitovirus 1, as well as plant mitochondria such as Anthurium amnicola, Solanum chacoense, Solanum tuberosum and Eucalyptus grandis, which clustered into one large clade belonging to the proposed *Mitoviridae* family (Figure 4A). BcMV10 is part of a separate branch clustered with fungi mitoviruses forming clade I. Mitoviruses from *B. cinerea* strains are widely distributed in Clades I, II and III. Based on Conserved Domain (CD) search results and multiple-sequence alignment analysis, BcMV10 RdRps contain typical conserved motifs (I–VI) which are characteristic of the mitoviral RdRp domain superfamily (pfam05919), similar to fungi mitoviruses, plant mitoviruses, and plant mitochondria to some extent (Figure 4B).

#### 3.4.2. Two Novel Botouliviruses

Phylogenetic tree analysis separated BcBoV18, BcBoV19 and related members into six groups in accordance with the genera *Botoulivirus*, *Penoulivirus*, *Scleroulivirus**, Magoulivirus*, *Rhizoulivirus* and *Ourmiavirus* in the *Botourmiaviridae* family. The sequence identities of BcBoV18 and BcBoV19 with members in the *Botourmiaviridae* family are in accordance with the phylogenetic tree results (Appendix A). BcBoV18 and BcBoV19 are divided into different branches in the *Botoulivirus* genus, which share 40% of their identity with RdRp aa sequences. Taken separately, BcBoV18 with BcOLV17 share high similarity, while BcBoV19, BcOLV12 and Sclerotinia minor botoulivirus 1 (SmBV1) are more closely related. However, mycoviruses from *B. cinerea* strains were divided into the genera *Penoulivirus* and *Magoulivirus*, except for *Botoulivirus*, revealing the presence of diverse botourmiaviruses in the *B. cinerea* natural population (Figure 5A). Based on a CD search of the NCBI database, putative conserved domains were detected in BcBoV18, while no putative conserved domains were detected in BcBoV19. Nevertheless, we tried to predict putative conserved domains in BcBoV19 using the MOTIF search tool (https://www.genome.jp/tools/motif, accessed on 28 July 2021). Multiple alignments of putative RdRp regions showed that BcBoV18, BcBoV19 and members of the family *Botourmiaviridae* contain eight conserved motifs (I to VIII), including a highly conserved GDD signature (Figure 5B). Therefore, BcBoV18 and BcBoV19 are two novel members of the *Botoulivirus* genus of the *Botourmiaviridae* family. Interestingly, nuclear localisation signals (NLSs) in RdRps of BcBoV18 and BcBoV19 were predicated using the online cNLS Mapper (Appendix A).

### 3.5. Distribution of Mycovirus-Derived RNA Reads along the Viral Genome

The RNA sequencing (RNA-seq) library was constructed and sequenced from the mixture samples. Mapping of RNA reads obtained from the RNA-seq library to the full-length genomes showed that 1,411,244 (3.42% overall alignment rate), 143,869 (0.035%) and 131,690 (0.32%) reads aligned with the obtained BcMV10, BcBoV18 and BcBoV19 genomic sequences, respectively. The RNA reads almost covered the full-length BcMV10, BcBoV18 and BcBoV19 RNAs. Much greater RNA read depth was achieved for the 3′- UTRs in the full-length genomes of BcBoV18 and BcBoV19. By contrast, no apparent read peaks were observed throughout the whole BcMV10 genome (Figure 6).

### 3.6. Horizontal Transmission of Mycoviruses

Pairing culture experiments between strains SZ-2-3y and B05.10 (SZ-2-3y/B05.10) were carried out (Figure 7A). Single-strain cultures of SZ-2-3y and B05.10 served as controls. Mycelial derivative isolates were obtained from colonies of B05.10 as recipients, and these (Yb-1, Yb-2, Yb-3, Yb-7, Yb-8 and Yb-11) grew almost the same as the B05.10 parental isolate (Figure 7B). When the tobacco leaves were inoculated with the three derivative isolates, they exhibited virulence similar to that of the B05.10 strain (Figure 7C). RT-PCR analysis showed that only BcBoV19 in Yb-3, Yb-8, and mixtures of BcBoV18, BcBoV19 and BcFV8 in Yb-7, were positive, while negative detection was observed for Yb-1, Yb-2 and Yb-11 from B05.10 derivatives (Table 2). Hyphal contact transmission experiments suggested that BcBoV19 alone, or the combination of BcBoV18, BcBoV19 and BcFV8 mycoviruses, could be successfully horizontally introduced into B05.10 with no apparent effect on growth, colony morphology or pathogenicity. However, the derivate with BcMV10 could not be obtained in multiple experimental replicates.

### 3.7. The BcMV10 Sequence Is Present in Cucumber Plants

#### 3.7.1. BcMV10 Sequences Are Present in Cucumber Plants Inoculated with Strain SZ-2-3y Mycelia

We inoculated SZ-2-3y mycelium agar onto cotyledons of cucumber plants. No SZ-2-3y mycelia were isolated from either the leaves or roots of cucumber plants (the data is not shown). Following RNA extraction from leaves or roots, nested RT-PCR, using different sets of BcMV10-based primers, yielded amplicons of expected sizes, suggesting that the BcMV10 sequence was present in cucumber plants. By contrast, no BcMV10-specific amplicons were obtained from healthy leaf tRNAs (Figure 8A, Appendix A). Associated experiments were conducted and (nested) RT-PCR analysis consistently confirmed the presence of the BcMV10 sequence in systemic leaves of ‘Cuiyu’ at 2 weeks post-inoculation. The average infection efficiency rate was ~50% (1/2) in the local leaves; 50% (2/4) and 33% (3/9) in the new leaves; and 2/4 (50%) in the roots. Meanwhile, it was positive for 10% (1/9) of the new leaves from inoculated ‘Suyo’ plants (Table 3). The presence of the BcMV10 sequence was also verified by Sanger sequencing of the above, randomly selected RT-PCR amplicons.

#### 3.7.2. BcMV10 Sequences Are Present in Cucumber Plants following Agroinfection with pXT1/BcMV10

The full-length BcMV10 cDNA clone and pXT1 were incorporated by homologous recombination [54]. Briefly, total RNA from the SZ-2-3y isolate was extracted and reverse-transcribed into cDNA to obtain the template. The full-length BcMV10 genome of 2975 nt was obtained and purified, and inserted into the linearised pXT1 expression vector. They were assembled to generate BcMV10 by transformation of *E. coli* and Agrobacterium cells. PCR detected 560 bp, 825 bp and 3522 bp target fragments that were used to screen positive clones from the above plasmids, which were designated pXT1/BcMV10 (Appendix A, Figure 8B). All positive clones were verified by DNA sequencing.

*Cucumis sativus* ‘Suyo’ and ‘Cuiyu’ plants were agroinoculated into the abaxial intercellular leaf space of two cotyledons with infectious clones of pXT1/BcMV10. An empty pXT1 vector served as a negative control (Figure 8B, Appendix A). At nearly 2 weeks post-inoculation, no obvious symptoms were observed in the systemic leaves of pXT1/BcMV10-inoculated plants, compared with negative controls. A (nested) RT-PCR analysis revealed the presence of the BcMV10 sequence in new leaves of pXT1/BcMV10-inoculated cucumber plants, with an average infection efficiency rate of 75% (3/4) and 22% (2/9) for ‘Cuiyu’, and 100% (8/8) for ‘Suyo’ (Table 3). These results suggest that the BcMV10 sequence was present in the new leaves of inoculated cucumber shoots.

## 4. Discussion

Currently, grey mold disease caused by *Botrytis* species threatens the horticultural crop, the *P. polyphylla* plant, by diminishing both its economic and medical value. *Botrytis* species can infect a large variety of hosts, including field-crop plants, fruit trees, vegetables and flowers [27,30,39,40]. Grey mold disease is a serious problem for agriculture worldwide, and current control relies on the excessive use of fungicides [27,28,29,30,40]. Unfortunately, cultivars/varieties that possess a high resistance to *Botrytis* species infection are not yet available. Nevertheless, mycovirus resources might be potential biocontrol agents for the control of fungal disease, because various mycoviruses have been identified and successfully used for chestnut blight disease and sclerotium disease [25,58,59,60,61,62].

In the present study, four mycoviruses (BcMV10, BcBoV18, BcBoV19 and BcFV8) were identified in the *B. cinerea* SZ-2-3y strain, the causal agent of *P. polyphylla* grey mold disease (Figure 1, Table 1). Based on genome structure, sequence homology, and phylogenetic analysis, all four are novel mycoviruses belonging to *Mitovirus*, *Botoulivirus* and *Fusarivirus* genera, respectively (Figure 2, Figure 3, Figure 4 and Figure 5). However, RdRp aa reveals pairwise identities of BcMV10, with the six closest mitoviruses clustering in the same branch ranging from 54% to 32% (Figure 4, Appendix A). These results revealed that mitoviruses are widely distributed and ancestrally present in fungal hosts.

Mycoviruses infecting a plant pathogenic *Botrytis* species are classified in different genera of *Botourmiaviridae*. Herein, BcBoV18 and BcBoV19 are clustered in two separate branches in the genus *Botoulivirus* (Figure 5, Appendix A). The BcMV10 reads had no apparent mapping depth along the whole genome, but BcBoV18 and BcBoV19 had high mapping depth to RNA reads in the 3′- termini of the full-length genome (Figure 6). The distinguishing distribution profiles could also be implicated in the different viral expression strategies for *Botouliviruses* and *Mitoviruses.* In conclusion, we characterised two novel botouliviruses and a novel mitovirus from *B. cinerea* strain SZ-2-3y, and provide insights into their molecular diversity and evolutionary origins.

It is well known that co-infections can result in interactions between viruses and hosts, displayed as antagonism or synergism [63,64,65,66]. It will be interesting to see whether interactions exist between the co-infected mitovirus, fusarivirus and the two botouliviruses. Confrontation cultures of SZ-2-3y and B05.10 showed an absence of BcMV10 but not of BcBoV19, and the mixture of BcBoV18, BcBoV19 and BcFV8 did not affect the phenotype and virulence of B05.10 on tobacco (Figure 7, Table 2). This result indicates that BcBoV18, BcBoV19 and BcFV8 might transmit horizontally, and have no apparent influence on the growth and pathogenicity of *B. cinerea* strains. Thus, we speculate that BcMV10 might be related to the hypovirulent SZ-2-3y strain. It is reported that BcMV1, SsMV1/HC025, SsMV3/NZ1 and SsMV4/NZ1 are hypovirulent, but their effects on fungal phenotype and biological characterisation are poorly understood due to the difficulty in isolating and transmitting to virus-free strains, as well as restricting the host antiviral RNA silencing machinery [6,15,17,22]. To our knowledge, BcMV10 is the tenth reported mitovirus infecting *B. cinerea*, and it is predicted to be related to the hypovirulence of SZ-2-3y; however, this needs to be further explored. In the future, we will attempt to obtain virus-free and BcMV10 strains using protoplast regeneration, which could shed light on the direct impact of BcMV10 on host biology and pathogenicity.

Bioinformatic analyses showed that related ourmia-like virus and mitovirus sequences are present in different organisms, suggesting a wide prevalence of these types of viruses across kingdoms [23,67,68]. Mitovirus sequences may transfer between fungi and plants, likely via fungus-mediated HGT events [19,20]. However, there is no direct evidence that mitovirus sequences can be transferred between fungi and plants. In the present study, we used the BcMV10-encoded protein in BLASTx searches for the presence of BcMV10-like sequences in the TSA-nr database. We identified several significant matches, with identity ranging from 26% to 37% in Pseudo-nitzschia australis, *Solanum chacoense* and *Anthurium Amnicola* (Appendix A). Viral genome organisation and phylogenetic analyses of BcMV10 indicate that this virus is closely related to fungal mitoviruses. Together with plant mitoviruses and plant mitochondria, they form a monophyletic cluster and belong to the proposed *Mitoviridae* family (Figure 4), consistent with their descent from a common ancestor [17,19]. We wondered whether the BcMV10 genome sequence is transmitted to host plants via HGT. Therefore, we tried to artificially inoculate cucumber plants with BcMV10 in different ways. Interestingly, partial BcMV10 genome sequences were detected in local and new leaves, and even the roots of cucumber plants inoculated with strain SZ-2-3y mycelia or Agrobacteria harbouring pXT1/BcMV10, respectively (Figure 8, Table 3, Appendix A). In contrast, BcBoV18, BcBoV19 and BcFV8 sequences were not detected in plants inoculated with strain SZ-2-3y mycelia. This reveals that the *B. cinerea* mitoviral sequence may exhibit transboundary entry into plants. Nevertheless, how mitoviral sequences are transmitted into plants, and whether they may be related to genetic material exchange, needs to be explored in future research.

Recently, cross-kingdom existence in different organisms has been explored. Some experimental evidence indicates that some plant viruses can replicate in fungus cells. Cucumber mosaic virus (CMV) may infect the phytopathogenic fungus *Rhizoctonia solani* in nature [69]. Additionally, plant viroid transcripts may infect phytopathogenic fungi based on artificial plant viroid infectious clones [70]. It is known that virus and virus-like organisms transfer between plants and fungi. An insect can transmit mycoviruses, which could provide opportunities for horizontal virus transfer between different organisms [71]. Thus, infectious mycovirus CHV1 may infect plants with the help of tobacco mosaic virus (TMV) and movement protein (MP), suggesting that mycovirus may horizontally transfer between fungi and plants [72]. Our results demonstrate the entry of a partial *B. cinerea* mitovirus genome sequence into cucumber plants, and suggest that BcMV10 could be closely linked to the horizontal transmission of mitoviruses between plants and fungi. We tried to inoculate B05.10 mycelia onto the new leaves of cucumber positive for BcMV10. The new leaves exhibited symptoms at 2 days after inoculation. We further isolated B05.10 mycelia from diseased leaves, which were negative for BcMV10 (Appendix A). However, it still not known how mitoviral sequences transmit between fungi and plants, and if mitoviruses can infect plants and exert specific biological functions. We believe that the discovery of BcMV10 significantly contributes to our understanding of the complex relationship between fungi, mitoviruses and plants.

## 5. Conclusions

Herein, we describe *Botrytis cinerea* strain SZ-2-3y, the causal agent of *P. polyphylla* grey mold disease in China. A novel mitovirus, two novel botouliviruses and a novel fusarivirus, isolated from the hypovirulent *B. cinerea* SZ-2-3y strain, were identified and molecularly characterised. Based on phylogenetic tree analysis, mitoviruses infecting *B. cinerea* species cluster into different clades. Conversely, some mitoviruses isolated from different fungal species share relatively close relationships. These results imply that mitoviruses may have evolved before fungal hosts diverged. In addition, alignment of RdRp aa sequences showed that BcMV10 is closely related to members of fungal mitoviruses. Together with plant mitoviruses and plant mitochondria, they form a monophyletic cluster and belong to the proposed *Mitoviridae* family, revealing their descent from a common ancestor. The BcMV10 genome sequence is present in tissues of cucumber plants experimentally inoculated with strain SZ-2-3y mycelia and agrobacterium-mediated pXT1/BcMV10. This provides evidence and new insight into the occurrence of cross-kingdom transfer of mitovirus sequences from fungi to plants, and illuminates aspects of mycoviral diversity and evolution. Whether or not BcMV10 plays a significant role in the relationship between fungi, mitoviruses and plants remain to be explored.

## Figures and Tables

**Figure 1 viruses-14-00151-f001:**
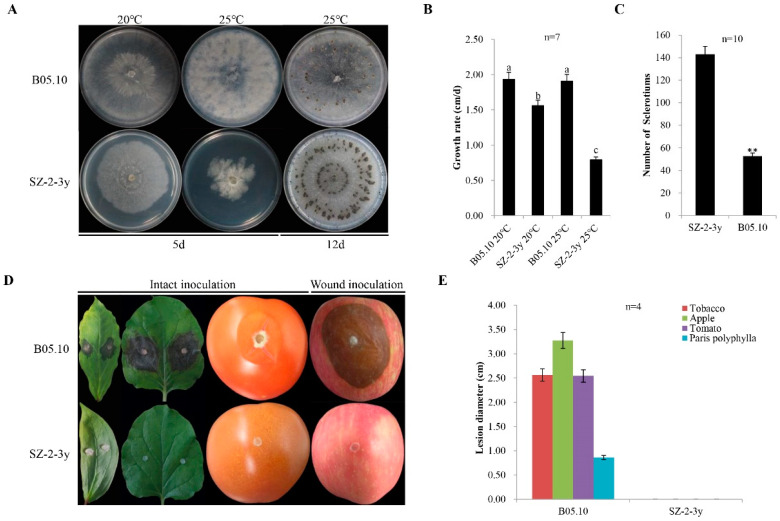
Culture characteristics and virulence of *Botrytis cinerea* isolates: (**A**) five-day-old cultures of the SZ-2-3y isolate and the B05.10 reference strain (20 °C and 25 °C) on potato dextrose agar (PDA), and production sclerotia at 20 °C for 12 days; (**B**) histogram showing average mycelial growth rates of the two isolates. Results are means ± standard deviation (SD; *n* = 7); (**C**) histogram showing the number of sclerotia for the two isolates (** *p* < 0.01); (**D**) pathogenicity of *B. cinerea* isolates on leaves of *Paris polyphylla* and tobacco, fruits of tomato following intact inoculation with mycelia, and apple fruits following wound inoculation with the mycelia using isolates of SZ-2-3y and B05.10, respectively (20 °C, 4 dpi); (**E**) histogram showing average lesion diameters of leaves of *P. polyphylla* and tobacco, and fruits of tomato and apple, caused by the two isolates. B05.10 served as a reference strain in the above experiments.

**Figure 2 viruses-14-00151-f002:**
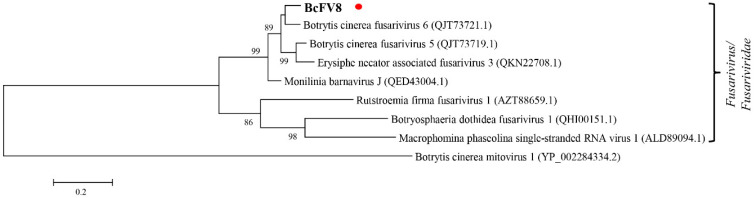
Phylogenetic tree analysis of the BcFV8 partial RdRp aa sequence with homologs from similar viruses. The BcFV8 identified in this study are represented in black body with red dot.

**Figure 3 viruses-14-00151-f003:**
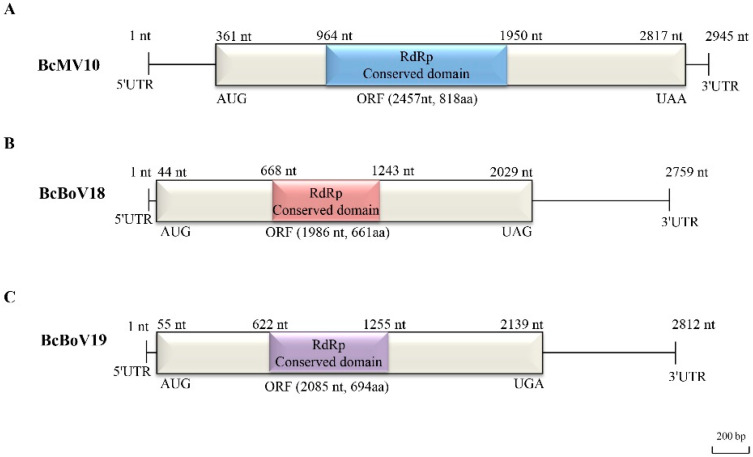
Full-length genome structures of: (**A**) BcMV10; (**B**) BcBoV18; and (**C**) BcBoV19. Each contains one open reading frame (ORF) flanked by two untranslated regions (UTRs) at 3′- and 5′- termini and RdRp conserved domain, which are indicated by an open bar and black lines, and different corresponding colour boxes, respectively.

**Figure 4 viruses-14-00151-f004:**
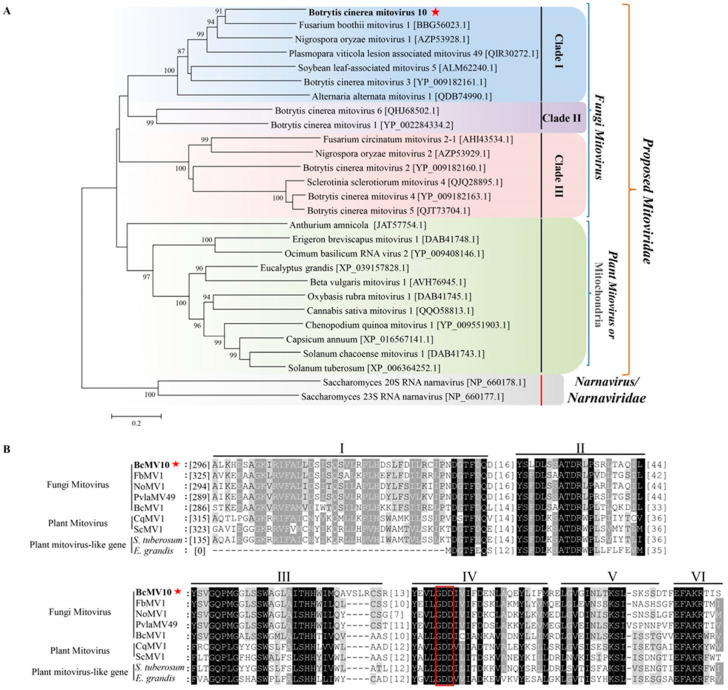
Neighbour-joining phylogenetic tree analysis: (**A**) multiple-sequence alignment comparisons of RdRp aa sequences of; (**B**) BcMV10, selected fungi mitoviruses, plant mitoviruses, and plant mitochondria belonging to the proposed *Mitoviridae* family. Mycoviral names, followed by GenBank accession numbers, are shown. Bold letters with red pentagrams represent the target virus BcMV10 characterised in this study. The conserved RdRp motifs of BcMV10 are indicated using Roman numerals from I to VI. The red box indicated the conserved domain GDD (motif IV).

**Figure 5 viruses-14-00151-f005:**
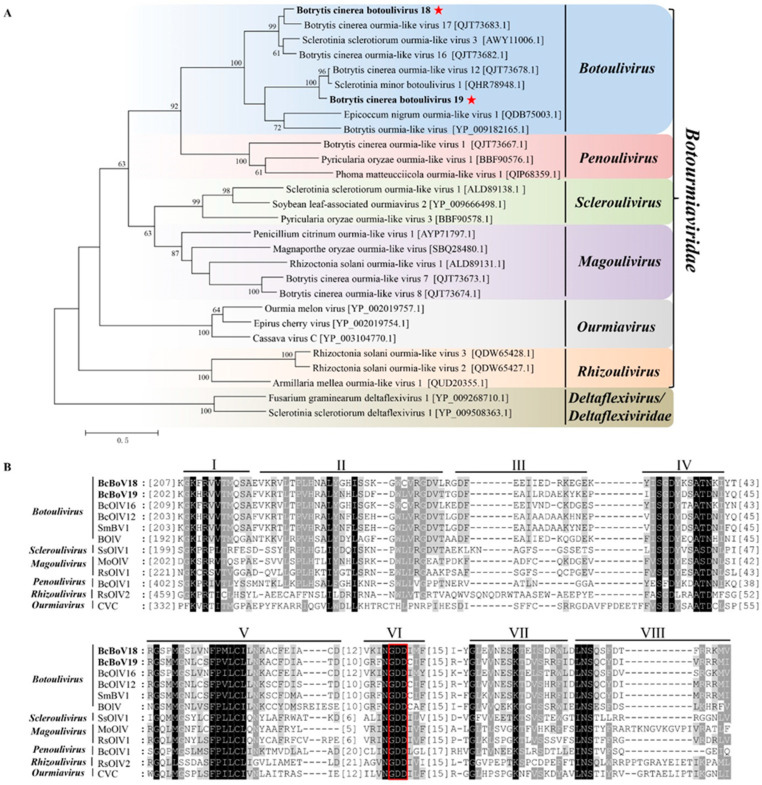
Phylogenetic tree depicting the relationships and: (**A**) multiple alignments of predicted RdRp aa sequences from; Botrytis cinerea botoulivirus 18, Botrytis cinerea botoulivirus 19 (indicated in the bold letters with red pentagrams)_are clustered into the genus *Botoulivirus*. (**B**) BcBoV18 and BcBoV19 (indicated with bold letters) in this study with members of *Botoulivirus*, *Penoulivirus*, *Scleroulivirus*, *Magoulivirus*, *Ourmavirus* and *Rhizoulivirus* in the *Botourmiaviridae* family. The conserved RdRp motifs are indicated using Roman numerals from I to VIII for BcBoV18 and BcBoV19. The red box indicated the conserved GDD motifs.

**Figure 6 viruses-14-00151-f006:**
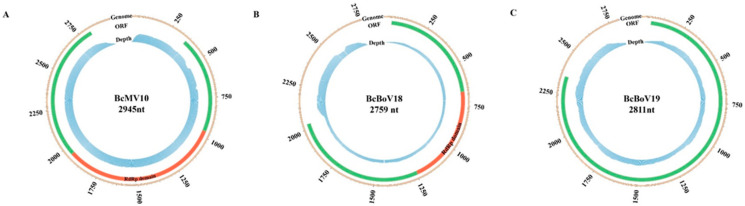
Profile distribution of viral reads along the full-length genomes of BcMV10, BcBoV18 and BcBoV19. Genome sequences (indicated with light brown colours), ORFs (indicated with light green colours), RdRp domains (indicated with red colours) and RNA read depth (indicated with light blue colours) of: (**A**) BcMV10; (**B**) BcBoV18; and (**C**) BcBoV19 are shown as circle graphs with different corresponding colours.

**Figure 7 viruses-14-00151-f007:**
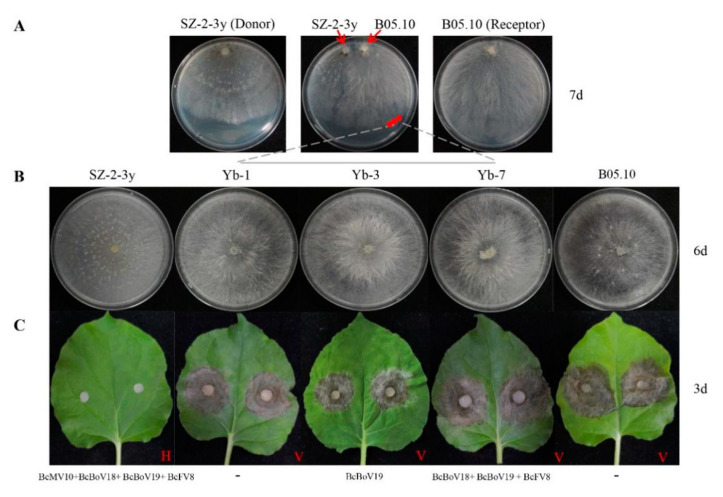
Horizontal transmission of mycoviruses through hyphal contact in paired cultures on PDA: (**A**) paired cultures after 7 days on PDA with colonies of strain SZ-2-3y as donor and strain B05.10 as receipt. Red circles indicate the area where a mycelial agar plug was removed and transferred to PDA for derivatives (Yb-1, Yb-3 and Yb-7); (**B**) colony phenotypes of the three representative derivatives on PDA (20 °C) after 6 days; (**C**) pathogenicity of derivatives (Yb-1, Yb-3 and Yb-7) on detached leaves of tobacco (20 °C, 3 days). H = Hypovirulence; V = Virulence. ‘-’ indicates no mycovirus detection.

**Figure 8 viruses-14-00151-f008:**
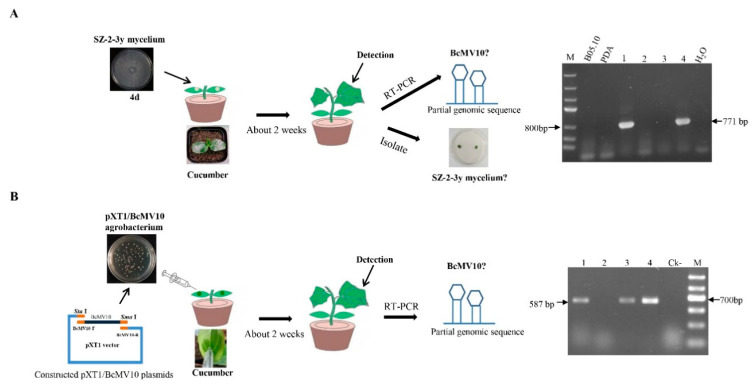
Schematic diagram of wound inoculation of cotyledons of cucumber plants with SZ-2-3y and RT-PCR detection of the BcMV10 sequence in new leaves of ‘Cuiyu’ plants inoculated with: (**A**) SZ-2-3y mycelia (M = Marker III, B05.10 mycelia and blank PDA-inoculated plants as negative controls; (**B**) agro-infiltration with the pXT1/BcMV10 clone (Ck- = pXT1-inoculated plants as negative controls; M-Marker II; Lanes 1–4, new leaves from four ‘Cuiyu’ cv. cucumber plants inoculated with SZ-2-3y mycelia (**A**) and pXT1/BcMV10 (**B**), respectively.

**Table 1 viruses-14-00151-t001:** Information for the assembled contigs of mycoviruses obtained from high-throughput sequencing analysis of *the Botrytis cinerea* SZ-2-3y strain.

Name	Contig ID	Contig Length (bp)	Protein	Best Match	GenBank Acc. No	Identity (%)	Query Coverage (%)	Genus/Family
Botrytis cinerea mitovirus 10	Contig111	2992	RdRp	Fusarium boothii mitovirus 1	BBG56024.1	54 (297/548)	77 (2305/2992)	*Mitovirus/Mitoviridae*
Botrytis cinerea botoulivirus 18	Contig334	2777	RdRp	Sclerotinia sclerotiorum ourmia-like virus 13	QUE49117.1	79 (528/668)	72 (1994/2777)	*Botoulivirus/Botourmiaviridae*
Botrytis cinerea botoulivirus 19	Contig420	2802	RdRp	Botrytis cinerea ourmia-like virus 12	QJT73678.1	79 (537/680)	72 (2036/2802)	
Botrytis cinerea fusarivirus 8	Contig12923	333	RdRp	Botrytis cinerea fusarivirus 5	QJT73719.1	85 (93/110)	99 (329/333)	*Fusarivirus/Fusariviridae*
	Contig25026	297				87 (85/98)	99 (293/297)	
	Contig46148	344				86 (31/36)	31 (107/344)	

**Table 2 viruses-14-00151-t002:** Horizontal transmission of mycoviruses from SZ-2-3y to B05.10.

Strains	Growth Rate (cm/day)	Pathogenicity Test	Mycoviruses
SZ-2-3y	1.48b	HV	BcMV10 + BcBoV18 + BcBoV19 + BcFV8
Yb-1	2.38a	V	Virus-free
Yb-2	2.35a	V	Virus-free
Yb-3	2.34a	V	BcBoV19
Yb-7	2.29a	V	BcBoV18 + BcBoV19 + BcFV8
Yb-8	2.34a	V	BcBoV19
Yb-11	2.32a	V	Virus-free
B05.10	2.38a	V	Virus-free

V = virulent; HV = hypovirulent.

**Table 3 viruses-14-00151-t003:** BcMV10 sequences were detected in cucumber plants following inoculation of cucumber cotyledons with SZ-2-3y mycelia and agro-infiltration with the pXT1/BcMV10 clone.

Experiment	Hosts	Replicates	Positive Rate by (Nested) RT-PCR			PCR Fragments
			Local Leaves	New Leaves	Roots	
SZ-2-3y mycelium	‘Cuiyu’ cv	1	1/2 (50%)	2/4 (50%)	2/4 (50%)	437 bp, 771 bp, 1721 bp
		2	/	3/9 (33%)	/	437 bp
	‘Suyo’ cv	3	/	1/9 (11%)	/	771 bp
BcMV10 infectious clone	‘Cuiyu’ cv	1	/	3/4 (75%)	/	587 bp
		2		2/9 (20%)		587 bp
	‘Suyo’ cv	3	/	8/8 (100%)	/	437 bp

‘/’ indicates no detection.

## Data Availability

The whole BcMV10, BcBoV18 and BcBoV19 genome sequences have been deposited in GenBank under accession numbers OK634394–6, respectively. The partial BcFV8 genome sequence has been deposited under GenBank accession number OL321741. Other data are available in Appendix A.

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
