# Peer review of "Four Novel Mycoviruses from the Hypovirulent Botrytis cinerea SZ-2-3y Isolate from Paris polyphylla: Molecular Characterisation and Mitoviral Sequence Transboundary Entry into Plants"

_viruses, 2022, doi:10.3390/v14010151_

Round 1
Reviewer 1 Report
This manuscript titled "Four novel mycoviruses founded in the hypovirulent Botrytis 2 cinerea SZ-2-3y isolate from Paris polyphylla: Molecular characterization and mitoviral sequence transboundary entry into plants" demonstrated the co-infection of four viruses in a hypovirulent strain of Botrytis cinerea. They determined that three viruses are not the cause of hypovirulence by horizontal transmission. They also found that the mitovirus BcMV10 can replicate in plants in the absence of the fungus. This reviewer wonders why they did not transform the fungal protoplasts using the same PXT1/BcMV10 construct using Agrobacterium tumefaciens-mediated transformation (ATMT). The same construct resulted in the replication in plants suggests that it should replicate in fungal mycelial fragments too. ATMT has been reported in the literature to be able used to transform fungal mycelial fragments (Dr. Weidong Chen at USDA-ARS). By adding the results from this batch of the experiment will provide definitive answer to the question whether BcMV10 causes hypovirulence.
Author Response
Response to Reviewer 1 Comments
A point-by-point response to the comments from the reviewer
Reviewer #1 comments and suggestions:
Point 1: This manuscript titled "Four novel mycoviruses founded in the hypovirulent Botrytis 2 cinerea SZ-2-3y isolate from Paris polyphylla: Molecular characterization and mitoviral sequence transboundary entry into plants" demonstrated the co-infection of four viruses in a hypovirulent strain of Botrytis cinerea. They determined that three viruses are not the cause of hypovirulence by horizontal transmission. They also found that the mitovirus BcMV10 can replicate in plants in the absence of the fungus. This reviewer wonders why they did not transform the fungal protoplasts using the same PXT1/BcMV10 construct using Agrobacterium tumefaciens-mediated transformation (ATMT). The same construct resulted in the replication in plants suggests that it should replicate in fungal mycelial fragments too. ATMT has been reported in the literature to be able used to transform fungal mycelial fragments (Dr. Weidong Chen at USDA-ARS). By adding the results from this batch of the experiment will provide definitive answer to the question whether BcMV10 causes hypovirulence.
Response 1: We greatly appreciate the positive and constructive comments from this reviewer.
Thank you for this insight suggestion. According to your good suggestions and ideas, we would like try to use pXT1/BcMV10 construct in combination with Agrobacterium tumefaciens-mediated transformation (ATMT) to transform the fungal protoplasts. As you suggested, agrobacterium tumefaciens-mediated transformation (ATMT) has been reported in the literature to be able used to transform fungal mycelial fragments with high efficiency (Dr. Weidong Chen at USDA-ARS). However, we noticed that the pXT1/BcMV10 recombinant vector in this study is a modified binary plant expression vector, which infects plant host. Therefore, we speculated that the pXT1/BcMV10 does not work well in fungi host.
We agree with these very helpful comments from the reviewer. We’ve recognized that some of the descriptions in the previous copy were not accurate and complete. The writings were checked and revised in English in the revised manuscript.

Reviewer 2 Report
Please see the report below and some errors or comments highlighted in the attachment.

Author Response
Response to Reviewer 2 Comments
A point-by-point response to the comments from the reviewer
Reviewer #2 comments and suggestions:
Point 1. Mitoviruses replicate exclusively in mitochondria. They used to be classified as Narnaviridae (ICTV 2009), but in ICTV 2020 they were labeled as a different family although closely related. Narnaviridae are the ones that do replicate in cytoplasm. Check https://academic.oup.com/ve/article/7/2/veab070/6352481 (Line 58 in the pdf version)
Response 1: Thank you for this insight suggestion. I checked mitoviruses replicate exclusively in mitochondria as you suggested. It was revised in the corresponding positions in the revised manuscript (Line 70 in the modified word version).
Point 2. This statement doesn't have any meaningful explanation of the statistical tests used to analyze the data. Normality, ANOVA? Duncan is just a posthoc method. (Line 140-143 in the pdf version)
Response 2: Thank you for this insight comment. A one-way analysis of variance (ANOVA) was used at a significance level of 0.05 with SPSS (version 22.0) to determine the differences between treatments. The sclerotia numbers and lesion diameters were analyzed using Student’s t test at p=0.05 or 0.01. It was revised in the corresponding positions in the revised manuscript as you suggested (Line 165-169 in the modified word version).
Point 3. Electrophoresis doesn't provide enough information about quality. Qubit RIN, 260/280, 260/230? (Line 150-151 in the pdf version)
Response 3: Thank you for this insight comment. Generally, the parameters of Qubit RIN, 260/280, 260/230 were used to assess the quality and quantity of dsRNA. The gel electrophoresis was used to observe and assess the dsRNA patterns. It was revised in the corresponding positions in the revised manuscript as you suggested (Line 178 in the modified word version).
Point 4. What for? There is no previous mention of this species before in the paper. (Line 155 in the pdf version); Then why mention it? (Line 162-163 in the pdf version); Why mention it? (Line 308-309 in the pdf version); No relevant to the present study (Line 433)
Response 4: Thank you for these insight comments. In total of 4 mycelium samples from three Alternaria spp. strains and SZ-2-3y in this study were together collected to be used for Illumina sequencing to screen mycovirus. As you referred above, three Alternaria spp. strains were not relevant to the present study. Therefore, it was not necessary to mention three Alternaria spp. strains in this study except for Illumina sequencings as mixture samples. It was revised in the corresponding positions in the revised manuscript as you suggested (Line 182-184, Line 196, Line 371, and Line 581 in the modified word version).
Point 5. Long-non coding RNA? What for? (Line 156 in the pdf version); Why long non-coding RNAs? (Line 432 in the pdf version)
Response 5: Thank you for these insight comments. The sequencing library was prepared from rRNA-depleted RNA. The description of LncRNA-seq is not accurate. It should be changed into ‘high-throughput sequencing’ instead of ‘LncRNA-seq’. It was revised in the corresponding positions in the revised manuscript as you suggested (Line 180, Line 186, and Line 581 in the modified word version).
Point 6. Details about analysis workflow are missing. Quality control, trimming, assembly method? (Line 159 in the pdf version)
Response 6: Thank you for these insight comments. The details about analysis of reads data as the followings. Clean data with high quality were obtained by filtering low quality from the raw data including of adapter-contaminated, unknown base (N) reads, and shorten reads. After a quality trimming step, de novo assembly was performed with IDBA_ud software. Subsequently, assembled contigs were further used to search for homology with mycoviral sequences using Blastn and BlastX in NCBI database. It was revised in the corresponding positions in the revised manuscript as you suggested (Line 189-194 in the modified word version).
Point 7. These statements don't belong here. They are more suited for a figure legend. (Line 188-192 in the pdf version)
Response 7: Thank you for your good suggestion. These statements were used to described corresponding the figure legend. It was revised in the corresponding positions in the revised manuscript as you suggested (Line 224, Line 520-523, Line 576-578 in the modified word version).
Point 8. Contradictory? The seeds were provided from CAAS or bought in online supermarkets? (Line 220-222 in the pdf version).
Response 8: Thank you for reminding us the inappropriate writing. The cucumber seeds of ‘Suyo’ cultivar (cv) were provided from Chinese Academy of Agricultural Science (CAAS), meanwhile the seeds of ‘Cuiyu’ cv were bought from online supermarkets. We have modified it in the corresponding positions in this revised manuscript according to your good suggestions (Line 258 in the modified word version).
Point 9. There is an extra "a" in the plot (Line 292 in the pdf version)
Response 9: Thank you for reminding us the inappropriate writing. The figure 1B is modified, which is not an extra "a" in the plot. The modified figure 1B was resubmitted in this revised manuscript.
Point 10. It should be p < 0.01(Line 294 in the pdf version)
Response 10: Thank you for reminding us the inappropriate writing. It should be p < 0.01 as you suggested. We have modified it in the corresponding positions in this revised manuscript (Line 353 in the modified word version).
Point 11. RdRp conserved domain shading is lacking in the figure. (Line 377 in the pdf version).
Response 11: Thank you for these insight comments. Bassd on the CD-search in NCBI, no putative conserved domains have been detected in BcBoV19. Therefore, RdRp conserved domain in BcBoV19 shading is lacking in the figure 3C. Nevertheless, we try to predict to obtain the putative conserved domains in BcBoV19 by online website of https://www.genome.jp/tools/motif. The RdRp conserved domain shading in BcBoV19 genome as the modified figure 3C was resubmitted in this revised manuscript.
We have modified it in the corresponding positions in this revised manuscript (Line 540-542 in the modified word version).
Point 12. Number? Criteria for selection? (Line 383 in the pdf version)
Response 12: Thank you for these insight comments. 25 and 24 representative members belonging to Mitoviridae and Botourmiaviridae were conducted, respectively. The criteria for selection were described in Section 3.4.1 and 3.4.2 in this manuscript, respectively. Briefly, the criteria for selection were described as the followings. 25 representative members belonging to Mitoviridae were selected based on significant matched sequences with mitovirus and mitochondria from the fungi and plant using the deduced BcMV10-encoded protein as the query. 24 representative members belonging to Botourmiaviridae were selected, which from the genus of Botoulivirus, Penoulivirus, Scleroulivirus, Magoulivirus, Rhizoulivirus and Ourmavirus in the Botourmiaviridae family exhibited significant matched identify for RdRp aa sequence alignments with BcBoV18 and BcBoV19. We have modified it in the corresponding positions in this revised manuscript (Line 471-473 in the modified word version).
Point 13. The descriptions especially marked with yellow color were not accurate and complete.
Response 13: We agree with your very helpful comments. We’ve recognized that some of the descriptions especially marked with yellow color in previous copy were not accurate and complete. We have modified it at the corresponding positions in this revised manuscript according to your good suggestions. The writings have been checked and revised in English in the revised manuscript.

Round 2
Reviewer 1 Report
- CaMV 35S promoter has been reported to work on driving fungal gene expressions. Even though the vector was originally intended for plant transformation, it is very likely to work in Botrytis system. For example: Establishment of Agrobacterium tumefaciens – mediated genetic transformation of apple pathogen Marssonina coronaria using marker genes under the control of CaMV 35S promoter, published in Microbiological Research.
- The experiment is also lacking information on whether the newly emerged leaves from BcMV10 agroinfiltrated plants became resistant to subsequent fungal inoculation compared to the controls. Please show the leaf assay because it could provide a strong evidence that BcMV10 plays a role in the cause of hypovirulence as the authors eluded, and yet are having trouble drawing a conclusion to.
Author Response
Response to Reviewer Comments
Point-by-point responses to reviewer comments:
Reviewer #1 comments and suggestions:
Point 1 CaMV 35S promoter has been reported to work on driving fungal gene expressions. Even though the vector was originally intended for plant transformation, it is very likely to work in Botrytis system. For example: Establishment of Agrobacterium tumefaciens–mediated genetic transformation of apple pathogen Marssonina coronaria using marker genes under the control of CaMV 35S promoter, published in Microbiological Research.
Response: Thank you for your insightful suggestion. We agree that the CaMV 35S promoter is likely to work in the Botrytis system. However, the system is not built at present. We will attempt to build the CaMV 35S promoter in the Botrytis host system in future research.
Point 2 The experiment is also lacking information on whether the newly emerged leaves from BcMV10 agroinfiltrated plants became resistant to subsequent fungal inoculation compared to the controls. Please show the leaf assay because it could provide strong evidence that BcMV10 plays a role in the cause of hypovirulence as the authors eluded, and yet are having trouble drawing a conclusion.
Response: We greatly appreciate these positive and constructive comments. We previously inoculated the B05.10 strain with virulent pathogenicity onto newly emerged leaves following exposure to SZ-2-3y mycelia, and the results were negative for BcMV10. The information was added to the Discussion section with Supplementary Figure S5. We have modified the corresponding positions in the revised manuscript (lines 843-846).
In our opinion, the manuscript mainly focused on the exploration of four newly discovered mycoviruses and their molecular characterisation in the SZ-2-3y host with hypovirulence that could serve as potential biocontrol agents. Intriguingly, a partial BcMV10 sequence was detected in cucumber plants. Our findings provide evidence of cross-kingdom mycoviral sequence transmission. Presently, we think we obtained these results and addressed the important scientific questions in the manuscript, which are significant.
We greatly appreciate the constructive insight, comments and helpful suggestions from you that have undoubtedly improved the manuscript. Again, thank you very much for your help. We hope the revised manuscript is sufficiently improved in both language and content.

Reviewer 2 Report
• English is of very poor quality. I have underlined in yellow the most obvious errors, ranging from lack of verbs or subjects to incomplete sentences or directly nonsensical. There are also some five or six line kilometer phrases, I have not marked those but it is something that should also be improved.
• The abstract has no structure. It lacks a justification, hypothesis and methods. As it is, only results count.
• The methods section is pretty poorly explained. Sometimes they give too much detail that hinders reading, and other times (generally in statistics and bioinformatics workflows) they give almost no information. In some sections it is not clear why they need the analysis they explain.
• The results read well, although the writing is sometimes unnecessarily intricate and makes it difficult to understand what they mean.
Author Response
Response to Reviewer Comments
Point-by-point responses to reviewer comments:
Reviewer #2 comments and suggestions:
English is of very poor quality. I have underlined in yellow the most obvious errors, ranging from lack of verbs or subjects to incomplete sentences or directly nonsensical. There are also some five- or six-line kilometer phrases, I have not marked those but it is something that should also be improved.
Response: Thank you for reminding us of these shortcomings. We agree with the helpful comments. We agree that some of the descriptions in the previous manuscript were not accurate or complete. We have modified the revised manuscript at the corresponding positions according to the valuable suggestions.
Point 1 The abstract has no structure. It lacks a justification, hypothesis and methods. As it is, only results count.
Response: Thank you for your insightful comments. The abstract has been revised, and the structure has been altered accordingly (lines 23-42 in the revised manuscript).
Point 2 The methods section is pretty poorly explained. Sometimes they give too much detail that hinders reading, and other times (generally in statistics and bioinformatics workflows) they give almost no information. In some sections it is not clear why they need the analysis they explain.
Response: The methods section has been improved according to the comments, including statistics and bioinformatics workflows, and some unnecessarily intricate detail has been removed. We hope the modified methods are more easily understood by the reader.
Point 3 The results read well, although the writing is sometimes unnecessarily intricate and makes it difficult to understand what they mean.
Response: The results section has also been improved according to the comments. We hope the modified results are more easily understood by the reader. The changes are highlighted in the revised manuscript.
We are grateful to the reviewer for the very helpful suggestions and comments. We have carefully checked and revised the manuscript following in line with the comments and suggestions. The revised manuscript has been checked by a native English-speaking colleague to correct language-related problems. Again, thank you very much for your help. We hope the revised manuscript is sufficiently improved in both language and content.
